# Correlation of Galectin Family Expression with Glioblastoma Progression and Survival

**DOI:** 10.3390/ijms27010417

**Published:** 2025-12-31

**Authors:** Peter Curpen, Farah Ahmady, Blaine M. H. Carnie, Grace E. C. Anderson, George Kannourakis, Amit Sharma, Adrian A. Achuthan, Rodney B. Luwor

**Affiliations:** 1Princess Alexandra Hospital, Brisbane, QLD 4102, Australia; jensencurpen@outlook.com; 2Fiona Elsey Cancer Research Institute, Ballarat, VIC 3350, Australia; farah@fecri.org.au (F.A.); blaine.carnie@student.unimelb.edu.au (B.M.H.C.); grace.anderson1@student.unimelb.edu.au (G.E.C.A.); george@fecri.org.au (G.K.); 3Federation University, Ballarat, VIC 3350, Australia; 4Department of Surgery, The University of Melbourne, The Royal Melbourne Hospital, Parkville, VIC 3050, Australia; 5Department of Integrated Oncology, Center for Integrated Oncology (CIO) Bonn, University Hospital Bonn, 53127 Bonn, Germany; amit.sharma@ukbonn.de; 6Department of Neurosurgery, University Hospital Bonn, 53127 Bonn, Germany; 7Department of Medicine, The University of Melbourne, The Royal Melbourne Hospital, Parkville, VIC 3050, Australia; aaa@unimelb.edu.au

**Keywords:** glioblastoma, galectins, tumour microenvironment, bioinformatics, immune evasion, prognostic biomarkers, therapeutic targets

## Abstract

Glioblastoma is the most aggressive primary brain malignancy, characterised by extensive intra-tumoural heterogeneity, therapy resistance, and a profoundly immunosuppressive tumour microenvironment. The galectin family, a group of β-galactoside-binding lectins, has emerged as a key regulator of tumour biology, influencing oncogenesis, immune modulation, and therapy resistance. In this study, we performed an integrative bioinformatics analysis to systematically evaluate the expression patterns, prognostic significance, genetic alterations, and functional roles of galectin family members in glioblastoma. We utilised publicly available genomic datasets and computational tools to perform our analysis, including UALCAN, GEPIA, cBioPortal, STRING, GeneMANIA, DAVID, and TIMER. We identified *LGALS1*, *LGALS3*, and *LGALS9* as significantly upregulated in glioblastoma, with their overexpression correlating with adverse patient survival. Functional enrichment analysis highlighted galectin-mediated pathways involved in extracellular matrix remodelling, immune dysregulation, tumour-promoting pathways, and protein processing, suggesting their pivotal role in glioblastoma pathogenesis. We also show that transcriptional and immunological signatures suggest that galectins may regulate glioblastoma immunosuppression, extracellular matrix remodelling, and protein homeostasis. Our findings provide novel insights into the oncogenic and immunoregulatory roles of galectins in glioblastoma, establishing their potential as prognostic biomarkers and therapeutic targets.

## 1. Introduction

Glioblastoma [formerly known as Glioblastoma Multiforme (GBM)] is the most aggressive primary malignant brain tumour in adults, classified as a World Health Organization (WHO) grade 4 glioma [1]. Even with current standard-of-care treatment, which includes maximal surgical resection combined with radiotherapy and temozolomide chemotherapy, prognosis remains extremely poor, with a median survival of only 12–18 months and a five-year survival rate of less than 7% [2,3]. The aggressive nature of glioblastoma is fuelled by marked intra-tumoral heterogeneity, therapy-resistant glioblastoma stem-like cells (GSCs), and a profoundly immunosuppressive tumour microenvironment [4,5]. Together, these features drive treatment failure and near-inevitable recurrence, underscoring the urgent need to identify novel molecular determinants of glioblastoma progression.

The emergence of large-scale and high-throughput genomic and transcriptomic datasets, such as those available through The Cancer Genome Atlas (TCGA), the Genotype-Tissue Expression (GTEx) project, and the Chinese Glioma Genome Atlas (CGGA), has enabled a comprehensive and systematic characterisation of the molecular landscape of glioblastoma. Leveraging these datasets through bio-informatics-driven approaches allows for the identification of novel prognostic biomarkers and provides insights into key molecular interactions that drive tumour progression. Among these, the galectin family—a group of β-galactoside-binding lectins—has attracted growing interest for its involvement in tumour progression, immune modulation, and therapeutic resistance [6,7].

Galectins influence a wide spectrum of key oncogenic processes, including cell adhesion, apoptosis resistance, angiogenesis, tumour inflammation and thrombosis, and extracellular matrix remodelling, making them attractive therapeutic targets in glioma and other solid tumours [6,7,8]. Beyond their classical functions as β-galactoside-binding lectins, galectins have emerged as key regulators of tumour–immune crosstalk, modulating T cell activation, immune checkpoint signalling, macrophage polarisation, and antigen presentation. In glioblastoma, mechanistic studies demonstrate that galectin activity intersects with several critical signalling axes, including Ras–MAPK-driven proliferation, NF-κB-mediated inflammation, hypoxia-induced HIF-1α responses, and TIM-3/PD-1-dependent immune evasion, highlighting their role in integrating extracellular cues with intracellular survival pathways. To date, Galectin-1 (encoded by the gene *LGALS1*), Galectin-2 (*LGALS2*), Galectin-3 (*LGALS3*), Galectin-4 (*LGALS4*), Galectin-7 (*LGALS7*), Galectin-8 (*LGALS8*), Galectin-9 (*LGALS9*), Galectin-10 (*LGALS10*/*CLC*), Galectin-12 (*LGALS12*), Galectin-13 (*LGALS13*), Galectin-14 (*LGALS14*), and Galectin-16 (*LGALS16*) have been identified in humans. Within this family, *LGALS1, LGALS3*, and *LGALS9* have been most consistently implicated in glioblastoma progression, through roles in tumour proliferation, immune evasion, and resistance to therapy [9,10,11,12,13].

Despite these insights, research to date has largely examined individual galectin members in isolation, and a comprehensive comparative analysis of the full galectin family in glioblastoma is lacking. This omission is notable given the structure similarity, overlapping ligand specificities, and potential functional redundancy within the galectin family. To address this critical gap, the present study undertakes a rigorous, integrative bioinformatics analysis of the galectin family in glioblastoma. Specifically, this study aims to (i) determine the differential expression profiles of galectin family members in glioblastoma relative to normal brain tissue, (ii) evaluate their prognostic significance through correlation with patient survival outcomes, and (iii) investigate their functional roles via pathway enrichment, gene co-expression analyses, and immune infiltration profiling.

## 2. Results

***LGALS1*, *LGALS3*, and *LGALS9* are overexpressed in glioblastoma patients.** The mRNA expression levels of galectin family genes in glioblastoma and normal brain tissues were first assessed using UALCAN. Compared to normal brain tissues, *LGALS1*, *LGALS3*, and *LGALS9* were significantly overexpressed in glioblastoma samples (Figure 1), with *LGALS9* exhibiting the highest transcript abundance. In contrast, *LGALS4* and *LGALS13* were markedly downregulated, suggesting a potential loss of function.

To validate these findings, differential expression was re-examined using GEPIA2 (Figure 2). Consistent with our UALCAN analysis, *LGALS1*, *LGALS3*, and *LGALS9* were also overexpressed in glioblastoma.

**Prognostic values of galectin family members in patients with glioblastoma.** To investigate the prognostic significance of galectin family members in glioblastoma, GEPIA2 was data-mined to assess a possible correlation between the mRNA expression levels of individual galectins with patient overall survival (OS) outcomes (Figure 3) and disease-free survival (DFS) outcomes (Figure 4). The analysis demonstrated that patients with low versus high levels of mRNA expression for each galectin family member assessed displayed no significant difference in overall survival (Figure 3).

However, high expression of *LGALS3* and *LGALS8* correlated with worse disease-free survival (DFS; Figure 4).

**Genomic alteration analysis of galectins in glioblastoma.** We next utilised the cBioPortal online platform to conduct a comprehensive genomic analysis of the galectin gene family in patients diagnosed with glioblastoma. The results delineated three principal categories of alterations that were identified and stratified by specific filtering criteria (Figure 5A), and the onco-print visualisation (Figure 5B) revealed a heterogeneous pattern of mutations, amplifications, and deletions across LGALS genes. Although alteration frequencies were low (<1% per gene), the diversity of alteration types suggests potential functional variation within the family.

Amplifications were most frequently observed in LGALS4, LGALS7, LGALS10, LGALS13, LGALS14, and LGALS16, whereas missense mutations were more common in LGALS8 and LGALS9. Deep deletions were detected across most galectin genes, except for LGALS8 and LGALS9. Truncating mutations and structural variants, though rare, were identified in LGALS13 and LGALS14. To evaluate the clinical significance of *LGALS* alterations, we examined their association with key survival metrics, including overall survival (OS) (Figure 5C), disease-free survival (DFS) (Figure 5D), progression-free survival (PFS) (Figure 5E), and disease-specific survival (DSS) (Figure 5F). No significant differences were observed across any survival metric.

**Construction of interactive gene network for galectin family members in glioblastoma**. GeneMANIA analysis identified 20 genes functionally associated with the galectin family, forming a core interaction network based on co-expression, shared protein domains, and genetic interactions (Figure 6A). To further characterise these relationships, the gene set was analysed using STRING, which generated a more detailed protein–protein interaction map comprising 32 seed genes, 52 nodes, and 226 edges (Figure 6B). The STRING network highlights extensive interconnectivity among galectins and their partners, suggesting coordinated involvement in key biological processes relevant to GBM pathogenesis. These interacting genes may therefore represent potential mediators—or therapeutic targets—within pathways governing GBM invasion, proliferation, and immune modulation.

**Functional enrichment analysis of co-expressed genes of galectins in glioblastoma.** To further determine the biological functions of galectins in glioblastoma, we employed the Database for Annotation, Visualization, and Integrated Discovery (DAVID) to perform Gene Ontology (GO) and Kyoto Encyclopedia of Genes and Genomes (KEGG) pathway analyses (Table 1). Functional enrichment analysis using the DAVID revealed that galectin-associated genes in GBM are primarily involved in pathways related to protein homeostasis and immune regulation.

GO biological process terms showed enrichment in protein folding, ER-associated degradation and suppression of CD4^+^ T cell proliferation, and type II interferon production, indicating coordinated roles in immune modulation and ER–Golgi quality-control mechanisms. Molecular function terms were dominated by carbohydrate- and galactoside-binding activities, alongside unfolded protein and complement C1q binding, consistent with the lectin biology of galectins. Cellular component enrichment highlighted localisation to the extracellular matrix, extracellular space, exosomes, and ER–Golgi intermediate compartments. KEGG analysis further identified significant enrichment in protein processing in the endoplasmic reticulum, antigen processing and presentation, and phagosome pathways, supporting a central role for galectin-related networks in protein quality control and immune signalling in GBM.

**The expression of galectin family members is correlated with immune infiltration levels in glioblastoma.** To investigate the role of galectins in the immune microenvironment of glioblastoma, we analysed their correlation with immune cell infiltration using the Tumor Immune Estimation Resource (TIMER) database (Figure 7). The partial correlation coefficients between each galectin family member and six major immune cell subsets (B cells, CD8^+^ T cells, CD4^+^ T cells, macrophages, neutrophils, and dendritic cells) are summarised in Table 2.

LGALS1 showed weak positive correlations with CD8^+^ T cells (r = 0.041), CD4^+^ T cells (r = 0.022), macrophages (r = 0.076), and neutrophils (r = 0.058), and a moderate association with dendritic cells (r = 0.433). LGALS3 demonstrated a similar pattern, with weak correlations across most immune subsets and a moderate positive association with dendritic cells (r = 0.462). LGALS8 also showed a moderate correlation with dendritic cell infiltration (r = 0.478), whereas its associations with other immune cells were weak.

LGALS9 exhibited the broadest pattern of positive correlations, including weak associations with macrophages (r = 0.137) and neutrophils (r = 0.406) and moderate correlations with CD4^+^ T cells (r = 0.313) and dendritic cells (r = 0.449). In contrast, LGALS13 displayed uniformly weak negative correlations with all immune cell types, suggesting a distinct immunological profile among the family members. Other galectins—including LGALS2, LGALS4, LGALS7, LGALS10/CLC, and LGALS14—showed only weak negative or weak positive correlations across all immune subsets, indicating limited involvement in immune infiltration based on transcriptomic data (Figure 7).

## 3. Discussion

The galectin family has attracted substantial interest due to its diverse roles in tumour progression, immune regulation, and cell signalling dynamics [14]. Galectins contribute to multiple malignant behaviours, including proliferation, invasion, metastasis, and immune evasion, and have been implicated in the progression of colorectal, breast, lung, and haematological cancers [15,16,17,18,19]. However, despite this expanding evidence base, their prognostic significance and mechanistic contribution within glioblastoma remains comparatively underexplored. In this study, we address this gap by applying an integrative bioinformatics approach to characterise the expression patterns, genomic alterations, prognostic relevance and associations, functional pathways, and immune-related interactions of galectins in glioblastoma.

Among the galectins analysed, *LGALS1*, *LGALS3*, and *LGALS9* emerged as significantly upregulated in glioblastoma, consistent with prior studies identifying these molecules as key contributors to glioma progression and suggesting multifaceted roles in tumour progression and resistance mechanisms. LGALS1, for instance, has been implicated in several hallmark features of glioblastoma pathogenesis, including resistance to chemoradiotherapy, induction of neo-angiogenesis, and suppression of anti-tumour immune responses [20]. It is internalised by endothelial cells, where it activates H-Ras signalling via the Raf/MEK/ERK cascade, ultimately driving endothelial cell proliferation and tumoural infiltration [20]. *LGALS3* is well-established as a mediator of glioma invasion, survival signalling, and therapeutic resistance, with its structural homology to Bcl-2 family proteins enabling anti-apoptotic activity through lactose-inhibitable interactions with Bcl-2 suppressing cytochrome C release and caspase activation, thereby enhancing chemo- and radioresistance [21]. Previous experimental work has also shown that *LGALS3* supports the maintenance of glioblastoma cancer stem cell (CSC) populations, further contributing to chemo- and radioresistance and tumour re-population [22,23]. Our observation of elevated *LGALS3* expression in glioblastoma is therefore concordant with these mechanistic findings and reinforces its biological relevance as a potential therapeutic target.

Complementing these tumour-intrinsic mechanisms, *LGALS9* appears to play a pivotal role in immune evasion and immunosuppression in the glioblastoma microenvironment, consistent with emerging evidence from recent immunogenomic studies. Recent studies showed that glioblastoma-derived exosomal *LGALS9* suppresses dendritic cell (DC) antigen presentation and cytotoxic T cell immunity in cerebrospinal fluid (CSF), providing a mechanistic basis for the profound immune dysfunction observed in glioblastoma [24,25]. These findings aligned well with our current findings that Galectin-9 expression and subsequent associations with immune cells may lead to overall favouring of immunosuppressive features in the glioblastoma microenvironment. By binding to TIM-3 on DCs, *LGALS9* disrupts antigen recognition, processing, and presentation, thereby impairing downstream cytotoxic T cell activation, a process that mirrors findings from functional experiments demonstrating TIM-3-dependent DC paralysis in glioma [24,25]. This mechanism aligns well with the established immunosuppressive microenvironment of glioblastoma, characterised by sparse effector T cell infiltration and an accumulation of immunosuppressive myeloid cells [24,25]. Interestingly, we have recently shown that peripheral T and NK cells from glioblastoma patients display greater levels of TIM-3 and lower activity markers CD69 and IFNγ compared to T and NK cells from healthy individuals [26]. Furthermore, a prior clinical study has shown that higher *LGALS9* expression in glioblastoma correlates with reduced survival outcomes, suggesting its potential as a prognostic biomarker [13]. Together, these data and our current findings underscore LGALS9 as a key regulator of immune escape in glioblastoma and reinforces its promise as a prognostic marker and potential therapeutic target.

In this study, GO and KEGG pathway enrichment analyses were undertaken to delineate the biological, molecular, and cellular functions associated with galectin activity in glioblastoma. The enrichment patterns highlight several key mechanisms through which galectins may influence tumorigenesis, including immune evasion, protein homeostasis, and ECM interactions. Notably, the significant enrichment of galectins in antigen processing and presentation pathways, phagosome formation, and the negative regulation of CD4-positive T cell proliferation implicates galectins as central modulators of the tumour–immune microenvironment. Our findings align with previous work demonstrating that *LGALS1*, *LGALS3*, and *LGALS9* bind to glycosylated ligands on T cells, inducing apoptotic cell death and attenuating anti-tumour immune responses [27].

Mechanistically, multiple galectins converge on apoptosis-inducing pathways in T cells through distinct yet complementary mechanisms. Galectin-1-triggered T cell apoptosis, for instance, is mediated through the activation of the JNK/c-Jun/AP-1 signalling pathway and the downregulation of the anti-apoptotic protein Bcl-2 [28,29]. Additionally, galectin-1 has been shown to interact with the extracellular domain of CD95 (APO-1/FAS) on resting T cells, promoting CD95 clustering to initiate caspase-8-dependent extrinsic T-cell apoptosis [30]. Galectin-3, acting extracellularly, binds glycosylated T cell ligands on the cell surface and activates the mitochondrial apoptotic pathway characterised by cytochrome C release and caspase-3 activation [31]. Galectin-9, through TIM-3 engagement, suppresses Th1 immune responses both directly by inducing TH1 apoptosis and indirectly by expanding immunosuppressive CD11b+ Ly-6G+ myeloid-derived suppressor cells (MDSCs) [32]. Collectively, the enrichment results from the present study extend these prior mechanistic observations by demonstrating that such immunoregulatory functions are reflected across a broader galectin-associated transcriptional network in GBM, rather than being restricted to isolated family members. The enrichment of galectin-associated genes within extracellular matrix (ECM)-related domains further supports a role in GBM invasiveness. The unique ECM composition of glioblastoma promotes angiogenesis, invasion, and therapeutic resistance of glioblastoma [33,34]. The enrichment of galectins in ECM-related components, including the collagen-containing extracellular matrix and extracellular exosomes, suggests a role in promoting tumour invasion and metastatic potential. Galectin-mediated mechanisms contribute to various tumour-promoting processes, including enhanced proliferation through oncogenic signalling pathways, evading growth suppressors, promoting inflammation, facilitating invasion via epithelial–mesenchymal transition (EMT), and early dissemination, inducing angiogenesis, enhancing resistance to cell death, and evading immune destruction [35,36,37,38,39]. Galectin-1, for instance, enhances glioblastoma cell migration via re-organisation of the actin cytoskeleton and upregulation of small GTPase RhoA expression [40,41,42]. Our data, demonstrating ECM enrichment, are consistent with these prior mechanistic findings and suggest that galectins collectively contribute to invasive phenotypes in glioblastoma.

Given the increasing emphasis on immunotherapeutic strategies for gliomas, understanding how galectins shape immune cell infiltration is clinically relevant in glioblastoma. In our TIMER analysis, distinct correlation patterns were observed across family members, emphasising the heterogeneous immunoregulatory functions of galectins in GBM. In particular, LGALS1, LGALS3, and LGALS9 demonstrated consistent positive associations with dendritic cells, macrophages, and CD8^+^ T cells, a pattern not observed in lower-expressed family members. These findings are consistent with previous studies showing that these three galectins play dominant roles in immune suppression and tumour progression in glioma. By contrast, LGALS13 was markedly negatively correlated with all immune cell types, suggesting a unique immunosuppressive profile that warrants further investigation.

Several limitations of this study should be acknowledged. First, all analyses were derived from publicly available transcriptomic and clinical datasets, such as TCGA and GEPIA2, which, although extensively curated, may be subject to inherent biases related to sample selection, batch effects, and platform-specific processing pipelines. These factors may influence gene expression estimates and limit the generalisability of our findings. Second, the absence of matched proteomic, epigenetic, and single-cell level data restricts our ability to validate transcriptional findings at the protein or cellular resolution. Consequently, the functional roles inferred from enrichment analyses remain hypothetical and require experimental confirmation.

Third, the clinical metadata available in these public datasets lack granularity regarding treatment history, molecular subtype (e.g., IDH status, MGMT methylation), extent of resection, and longitudinal disease course. These omissions constrain the precision of our prognostic analyses and preclude detailed stratified assessments that may reveal subtype-specific galectin functions. Finally, although our immunological analyses offer insight into potential galectin–immune interactions, computational deconvolution cannot fully capture the spatial and functional complexity of the glioblastoma immune microenvironment.

Future research should integrate multi-omics datasets—including proteomics, methylomics, metabolomics, and spatial transcriptomics—to validate and extend the transcriptional patterns identified here. In vitro and in vivo studies will be essential to define the mechanistic contributions of individual galectins to tumour progression, immune suppression, and therapeutic resistance. Large, well-annotated clinical cohorts and single-cell sequencing will further clarify the cell type-specific roles of galectins and their relevance across molecular glioma subtypes. Importantly, incorporating epigenomic profiling—such as DNA methylation and chromatin accessibility analyses—will help determine whether galectin dysregulation reflects underlying epigenetic programmes. Such multi-layered approaches will be crucial to establish whether individual galectins, or coordinated galectin networks, represent viable biomarkers or therapeutic targets in glioblastoma.

This study provides a comprehensive, family-wide analysis of galectins in glioblastoma, revealing that LGALS1, LGALS3, and LGALS9 are consistently overexpressed and strongly associated with adverse survival, immune dysregulation, and tumour-promoting pathways. Although genomic alterations were rare, transcriptional and immunological signatures suggest that galectins contribute to key processes underpinning glioblastoma aggressiveness, including immune suppression, protein homeostasis, and extracellular matrix remodelling. These findings establish a foundational framework for future mechanistic and translational studies and highlight galectins—individually and as a coordinated network—as promising candidates for biomarker development and therapeutic targeting in glioblastoma.

## 4. Materials and Methods

**UALCAN:** The interactive web portal UALCAN (http://ualcan.path.uab.edu) was utilised to comprehensively analyse the expression levels of the galectin family genes in tumour and normal specimens, leveraging data from The Cancer Genome Atlas (TCGA) database [43]. This platform enables the systematic investigation of gene expression differences, allowing for stratification across various tumour subgroups, including distinctions based on tumour stage, histological subtype, patient sex, and other clinicopathological parameters. Differential expression analyses are performed using TPM-normalised transcript abundance and Welch *t*-tests with *p* < 0.05 denoting statistical significance.

**GEPIA:** The expression profiles of galectin family genes in glioblastoma were analysed using the Gene Expression Profiling Interactive Analysis (GEPIA) database (http://gepia.cancer-pku.cn/) [44]. GEPIA is an online platform that allows for the comparison of gene expression levels between tumour and paired normal tissues by integrating data from The Cancer Genome Atlas (TCGA) and the Genotype-Tissue Expression (GTEx) database. To assess the prognostic significance of galectin gene expression in glioblastoma, patients were categorised into high-expression and low-expression cohorts using the median expression value (50%) as the threshold. Specifically, samples with expression levels above the median were assigned to the high-expression group, while those with expression levels below the median were classified as the low-expression group. Kaplan–Meier survival analysis was performed to evaluate overall survival (OS) and disease-free survival (DFS) based on galectin expression. The statistical significance of survival differences between high- and low-expression groups was assessed using the log-rank test. Additionally, the hazards ratio (HR) was calculated using the Cox proportional hazards (Cox PH) model to determine the association between galectin gene expression and patient survival outcomes in glioblastoma. Expression values were based on TPM normalisation and log@ transformation within the GEPIA pipeline.

**cBioPortal analysis:** The genetic alterations of galectin genes in glioblastoma were investigated using the cBio Cancer Genomics Portal (cBioPortal) (http://cbioportal.org), a comprehensive platform for the exploration of multidimensional cancer genomics data [45]. This analysis aimed to identify the frequency and types of mutations occurring in galectin genes within glioblastoma samples. Following the standard analytical framework of cBioPortal, the mutation landscape of galectin genes was assessed, including an overview of mutation types and their distribution within glioblastoma cases. Additionally, survival analyses were conducted to evaluate the prognostic impact of galectin mutations. OS and DFS were compared between patient cohorts with and without galectin mutations. Survival curves were generated using Kaplan–Meier methodology, and statistical significance was assessed using the log-rank test, with *p* < 0.05 denoting significance.

**GeneMANIA:** To examine the network of galectin family members and their associated genes, GeneMANIA (http://www.genemania.org) was utilised [46]. GeneMANIA is a widely employed bioinformatics tool that integrates multiple functional association networks, including co-expression, co-localization, genetic interactions, and shared protein domains, to predict gene function and connectivity. We utilised GeneMANIA to analyse the relationship between galectin family members and co-expression genes.

**STRING analysis:** STRING (https://string-db.org) was used to analyse protein–protein interaction networks among galectin family members and their 20 associated genes [47]. STRING integrates multiple sources of interaction evidence—including data from experimental studies, computational predictions, and current literature mining—to generate high-confidence functional interaction networks. Interaction maps were constructed using the confidence-based model, in which edge weights reflect interaction scores derived from aggregated evidence. A minimum interaction confidence threshold of >0.4 (medium confidence) was applied. The resulting network visualisation illustrated the interconnectivity between galectin family members and their co-expressed genes, highlighting key functional pathways and putative interaction partners.

**DAVID:** To determine the functional roles of galectin genes and their co-expressed genes, functional enrichment analysis was performed using the Database for Annotation, Visualization, and Integrated Discovery (DAVID) (https://davidbioinformatics.nih.gov/) [48]. The input gene list consisted of all galectin family members together with the top 20 co-expressed or functionally linked genes identified through GeneMANIA. Analyses were conducted using Homo sapiens as a background reference, and the Functional Annotation Tool was used to generate enrichment statistics across multiple annotation categories. Gene Ontology (GO) and Kyoto Encyclopedia of Genes and Genomes (KEGG) pathway enrichment analyses were performed to identify the molecular functions (MFs), biological processes (BPs), and cellular components (CCs) associated with galectin genes and their interactors. GO analysis provided insights into gene functions at the molecular, cellular, and organismal levels. KEGG pathway analysis was used to map the input gene set to canonical signalling pathways relevant to glioblastoma. KEGG terms were assessed to identify enriched immune-modulatory pathways and signalling intermediates and share molecular hubs among galectin family members. These analyses provided mechanistic context for the expression patterns observed in glioblastoma. A false discovery rate (FDR) threshold of <0.05 was applied to determine the statistical significance of enriched GO terms and KEGG pathways corrected using the Benjamini–Hochberg method. Enrichment results were ranked based on their significance, represented as the enrichment score (−log_10_(*p*-value)), to highlight the most relevant functional categories and pathways associated with galectin gene activity in glioblastoma.

**TIMER analysis:** To investigate the relationship between galectin expression and tumour-infiltrating immune cells in glioblastoma, we utilised TIMER (https://cistrome.shinyapps.io/timer/), a web-based bioinformatics platform designed for the comprehensive analysis of immune infiltration across multiple cancer types [49]. In this study, TIMER was used to evaluate the correlation between galectin expression levels and the infiltration of immune cell subsets within glioblastoma. The platform enables the quantification of key immune cell populations, including B cells, CD4^+^ T cells, CD8^+^ T cells, macrophages, neutrophils, and dendritic cells, facilitating an in-depth exploration of the immunological role of galectins in the tumour microenvironment—using a validated deconvolution algorithm applied to TCGA RNA-Seq data. Correlation analyses were conducted to determine associations between galectin expression and immune infiltration patterns. Specifically, correlations between galectin expression and immune cell infiltration were quantified using Spearman’s rank correlation co-efficient. Correlation strength was interpreted according to conventional thresholds (weak = R < 0.3; moderate = R 0.3–0.5; strong = R > 0.5). Statistical significance was determined using TIMER’s permutation-based testing framework, with a significance threshold of *p* < 0.05.

## 5. Conclusions

This study represents a large-scale, bioinformatics-driven characterisation of the galectin family in glioblastoma, providing novel insights into their expression patterns, prognostic significance, and potential therapeutic relevance. Our study highlights the role of *LGALS1*, *LGALS3*, and *LGALS9* in glioblastoma invasion and migration via the induction of T cell apoptosis, protein homeostasis, and the maintenance of the ECM. Future research should focus on mechanistic validation, translational drug development, and clinical evaluation of galectin-based therapeutic approaches, particularly in the context of immunotherapy. Given the profound challenges associated with treating glioblastoma, targeting the galectin family may represent a novel and promising avenue for improving patient outcomes in this devastating disease.

## Figures and Tables

**Figure 1 ijms-27-00417-f001:**
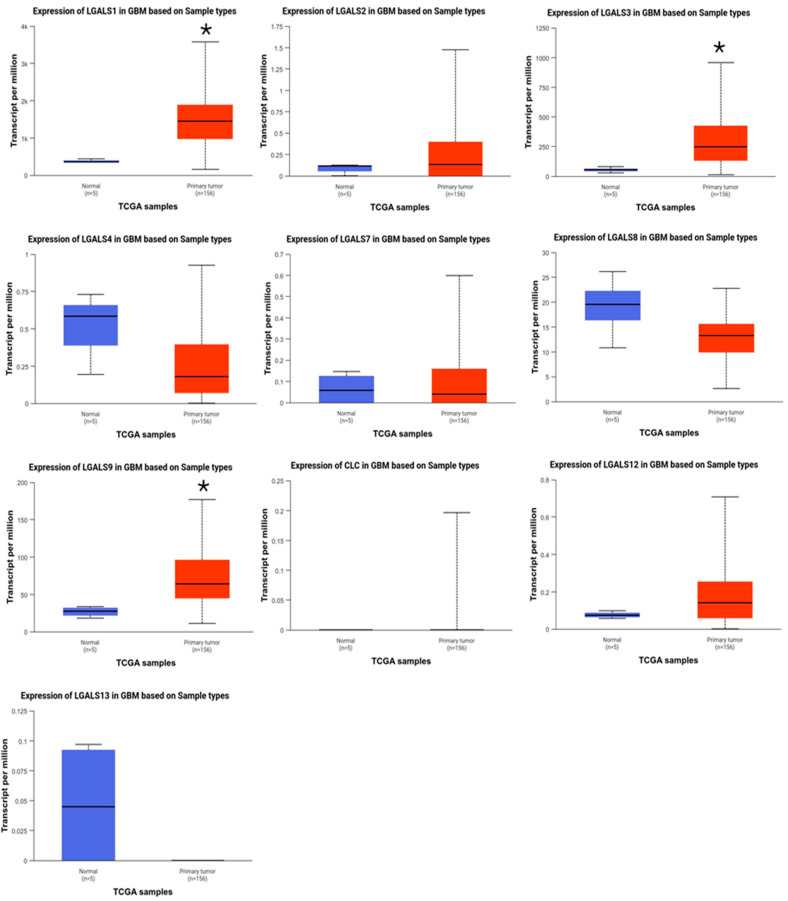
Comparison between the mRNA expression of galectins in glioblastoma versus normal tissue (UALCAN). Gene expression (transcripts per million) of normal brain tissue (blue; n = 5) and primary glioblastoma (red; n = 156) was examined for the human galectin family. LGALS1, LGALS3, and LGALS9 were significantly overexpressed in glioblastoma samples compared to normal brain tissue. No data was available for LGALS14 and LGALS16. * *p* ≤ 0.05.

**Figure 2 ijms-27-00417-f002:**
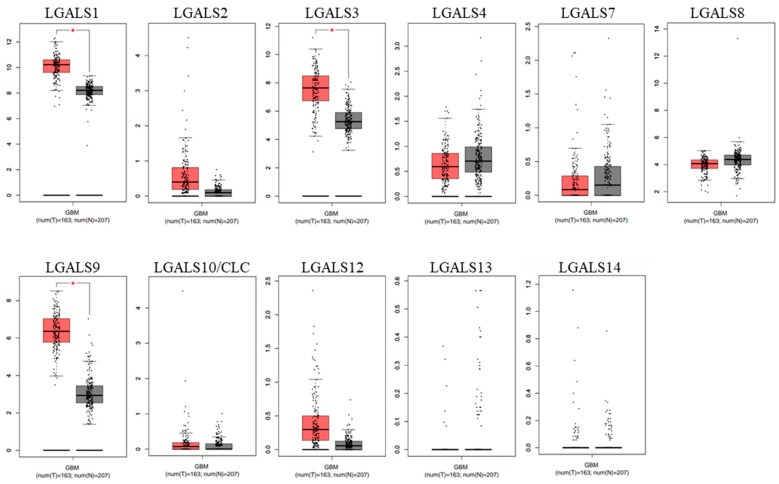
Comparison between the mRNA expression of galectins in glioblastoma versus normal tissue (GEPIA2). Gene expression of normal brain tissue (grey; n = 163) and primary glioblastoma (red; n = 207) were examined for the human galectin family. LGALS1, LGALS3, and LGALS9 were significantly overexpressed in glioblastoma samples compared to normal brain tissue. No data was available for LGALS16. * *p* ≤ 0.05.

**Figure 3 ijms-27-00417-f003:**
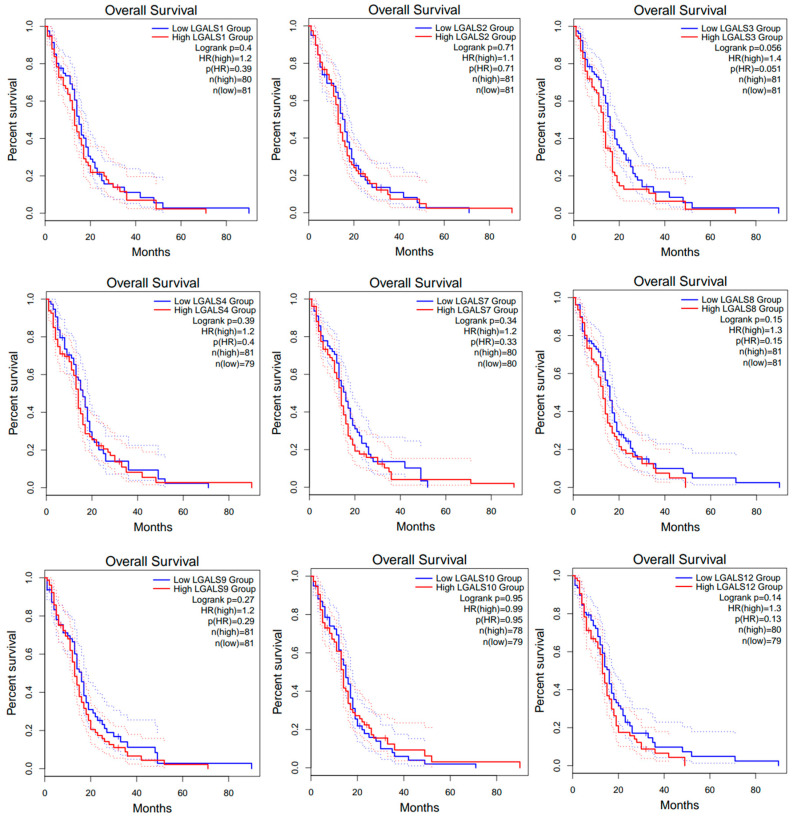
Overall survival (OS) curves for glioblastoma patients from the GEPIA2 database. Kaplan–Meier curves compare OS between high- (red line) and low-expression (blue line) groups for each LGALS gene, defined using the median expression cut-off. Dotted lines indicate the 95% confidence intervals for each curve and do not represent additional patient groups or independent survival curves. Log-rank *p* values and hazard rations (HRs) were computed using a COX proportional hazard model. No data was available for LGALS13, LGALS14 and LGALS16.

**Figure 4 ijms-27-00417-f004:**
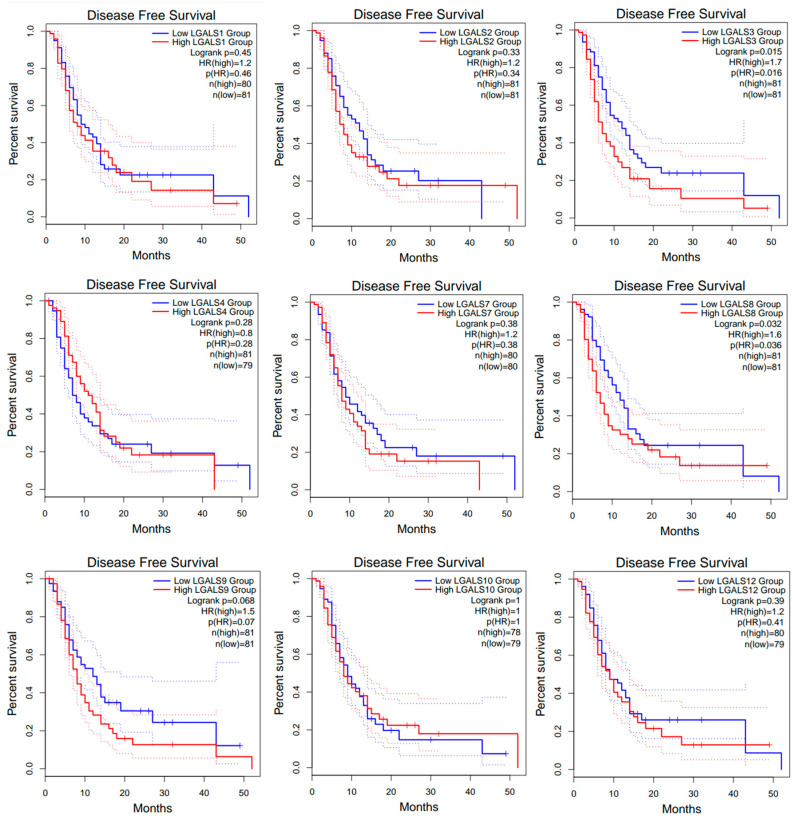
Disease-free survival (DFS) curves for glioblastoma patients from the GEPIA2 database. Kaplan–Meier curves compare DFS between high- (red line) and low-expression (blue line) groups for each LGALS gene, defined using the median expression cut-off. Dotted lines indicate the 95% confidence intervals for each curve and do not represent additional patient groups or independent survival curves. Log-rank *p* values and hazard rations (HRs) were computed using a COX proportional hazard model. No data was available for LGALS13, LGALS14, and LGALS16.

**Figure 5 ijms-27-00417-f005:**
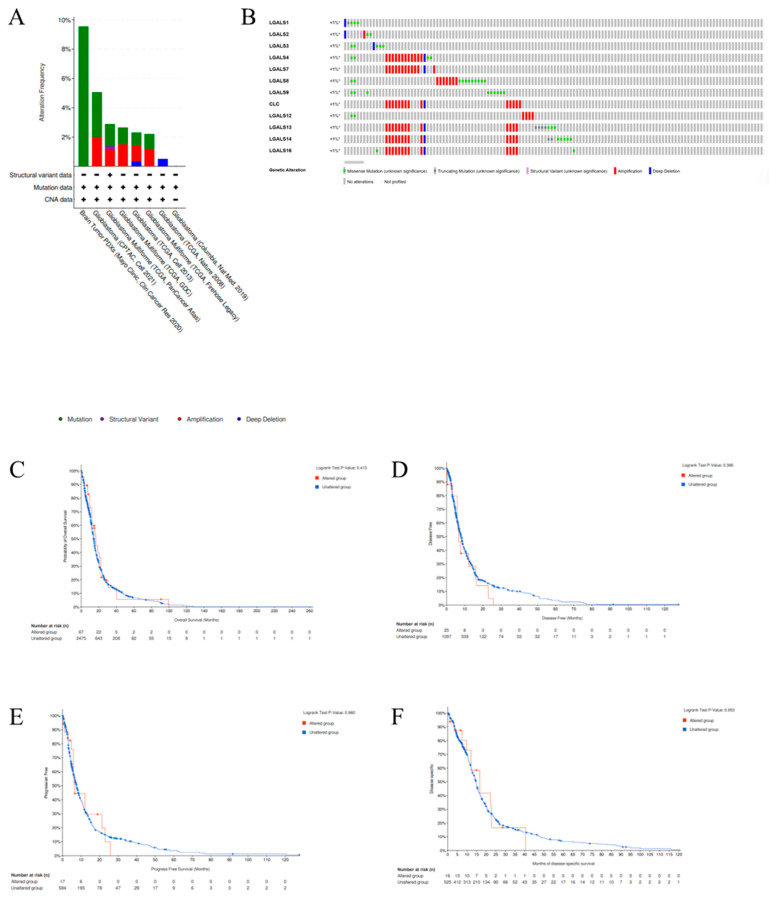
Alteration frequency of the galectin family genes in glioblastoma patients. (**A**). Summary of the cancer types in the cBioPortal used to calculate the percentages of glioblastoma cases with the galectin family genes. (**B**). mRNA expression alterations (RNA Seq V2 RSEM) of the galectin family genes in glioblastoma patients. (**C**). Overall survival (OS), (**D**). disease-free survival (DFS), (**E**). progress-free survival (PFS), and (**F**). disease-specific survival (DSS) of glioblastoma patients with altered (red line) and unaltered (blue line) mRNA expression of the galectin family genes.

**Figure 6 ijms-27-00417-f006:**
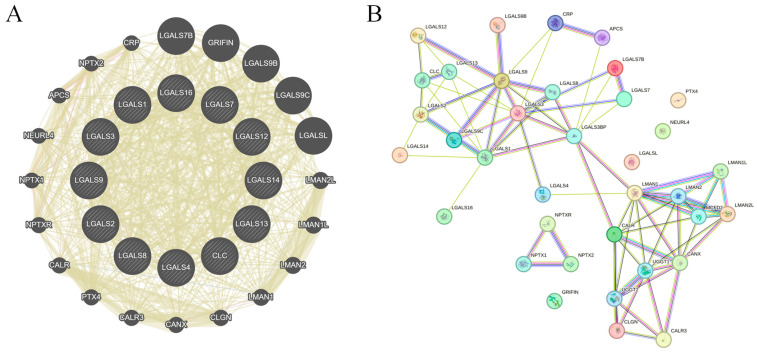
Gene–gene network of galectin family members in glioblastoma. (**A**). GeneMANIA-derived gene–gene interaction network showing functional associations among galectin family members and their top co-expressed genes. Node size reflects connectivity (degree), while edge width represents interaction strength. All edges are displayed using GeneMANIA’s default unified colour scheme and do not distinguish between specific interaction types. (**B**). STRING protein–protein interaction (PPI) network of galectin family members and correlated genes. Edges represent confidence-weighted interactions based on integrated evidence (experimental data, curated databases, co-expression, and text mining). Node colour reflects STRING clustering and does not indicate interaction type.

**Figure 7 ijms-27-00417-f007:**
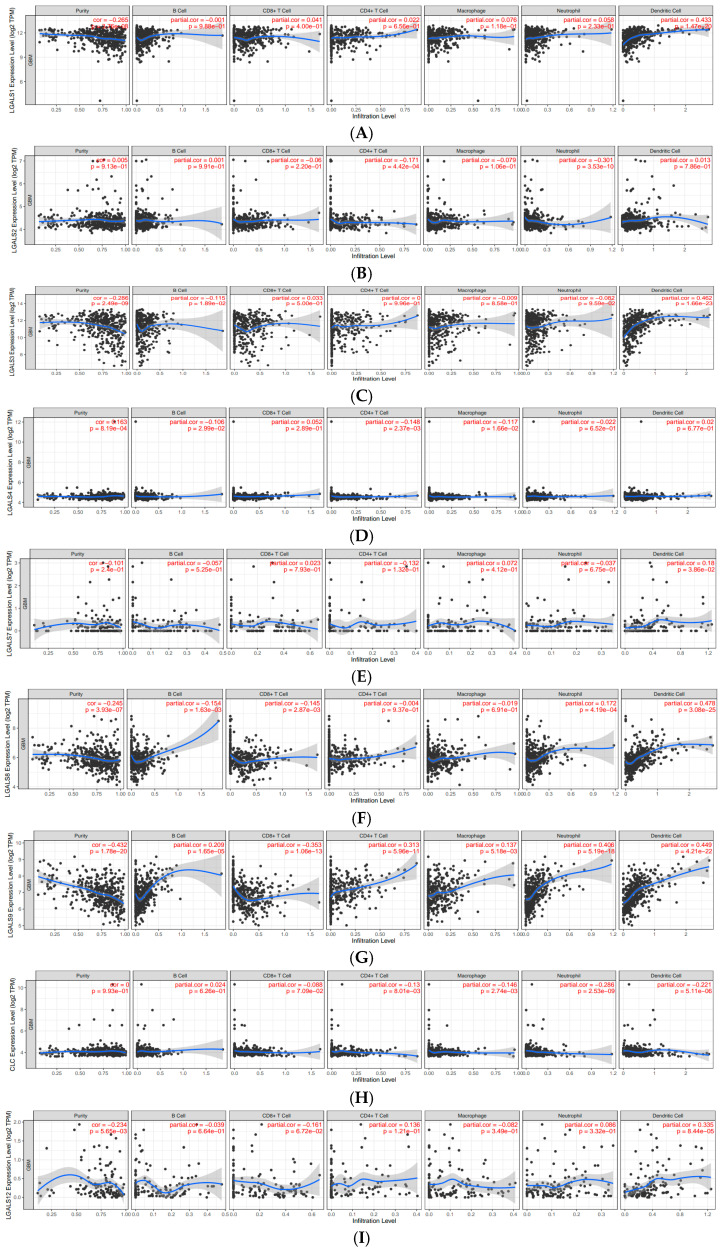
Correlations of galectin family gene expression with immune infiltration level in glioblastoma. The correlation between each type of TIICs (B-cells, CD4^+^ T cells, CD8^+^ T cells, neutrophils, macrophages, and dendritic cells) and (**A**). LGALS1, (**B**). LGALS2, (**C**). LGALS3, (**D**). LGALS4, (**E**). LGALS7, (**F**). LGALS8, (**G**). LGALS9, (**H**). LGALS10/CLC, (**I**). LGALS12, (**J**). LGALS13, and (**K**). LGALS14. No data was available for LGALS16.

**Table 1 ijms-27-00417-t001:** GO annotation and KEGG pathway enrichment analysis of galectin family members in glioblastoma.

Category	Term	Gene Count	%	*p*-Value
GOTERM_MF_DIRECT	Carbohydrate binding	23	74.2	8.00 × 10^−38^
GOTERM_MF_DIRECT	Galactoside binding	6	19.4	3.70 × 10^−13^
GOTERM_MF_DIRECT	Unfolded protein binding	6	19.4	1.60 × 10^−6^
GOTERM_MF_DIRECT	Lactose binding	3	9.7	7.10 × 10^−6^
GOTERM_MF_DIRECT	D-mannose binding	4	12.9	8.70 × 10^−6^
GOTERM_MF_DIRECT	Complement component C1q complex binding	3	9.7	1.30 × 10^−4^
GOTERM_BP_DIRECT	Protein folding	7	22.6	9.10 × 10^−8^
GOTERM_BP_DIRECT	Negative regulation of CD4-positive, alpha-beta T cell proliferation	4	12.9	6.00 × 10^−7^
GOTERM_BP_DIRECT	Negative regulation of type II interferon production	4	12.9	3.10 × 10^−5^
GOTERM_BP_DIRECT	ERAD pathway	4	12.9	2.80 × 10^−4^
GOTERM_BP_DIRECT	Positive regulation of gene expression	6	19.4	4.90 × 10^−4^
GOTERM_BP_DIRECT	Endoplasmic reticulum to Golgi vesicle-mediated transport	4	12.9	1.80 × 10^−2^
GOTERM_CC_DIRECT	Collagen-containing extracellular matrix	9	29	3.40 × 10^−8^
GOTERM_CC_DIRECT	COPII-coated ER to Golgi transport vesicle	4	12.9	2.50 × 10^−5^
GOTERM_CC_DIRECT	Extracellular space	11	35.5	1.70 × 10^−4^
GOTERM_CC_DIRECT	Endoplasmic reticulum–Golgi intermediate compartment	4	12.9	1.70 × 10^−4^
GOTERM_CC_DIRECT	Endoplasmic reticulum–Golgi intermediate compartment membrane	4	12.9	1.20 × 10^−4^
GOTERM_CC_DIRECT	Endoplasmic reticulum membrane	8	25.8	7.60 × 10^−3^
KEGG_PATHWAY	Protein processing in endoplasmic reticulum	5	16.1	1.30 × 10^−7^
KEGG_PATHWAY	Antigen processing and presentation	2	6.5	3.60 × 10^−2^
KEGG_PATHWAY	Phagosome	2	6.5	7.00 × 10^−2^
KEGG_PATHWAY	Human T cell leukemia virus 1 infection	2	6.5	9.70 × 10^−2^

GO, gene ontology; KEGG, Kyoto encyclopedia of genes and genomes; BP, biological process; CC, cellular component; MF, molecular function.

**Table 2 ijms-27-00417-t002:** Correlation of galectin family member expression with immune cell infiltration in glioblastoma.

Galectin Family	B Cell	CD8^+^ T Cell	CD4^+^ T Cell	Macrophage	Neutrophil	Dendritic Cell
LGALS1	−0.001	0.041	0.022	0.076	0.058	0.433
LGALS2	0.001	−0.06	−0.171	−0.079	−0.301	0.013
LGALS3	−0.115	0.033	0	−0.009	−0.082	0.462
LGALS4	−0.106	0.052	−0.148	−0.117	−0.022	0.02
LGALS7	−0.057	0.023	−0.132	0.072	−0.037	0.18
LGALS8	−0.154	−0.145	−0.004	−0.019	0.172	0.478
LGALS9	0.209	−0.353	0.313	0.137	0.406	0.449
CLC	0.024	−0.088	−0.13	−0.146	−0.286	−0.221
LGALS12	−0.039	−0.161	0.136	−0.082	0.086	0.335
LGALS13	0	−0.066	−0.079	−0.097	−0.079	−0.063
LGALS14	0.071	−0.09	−0.072	−0.046	−0.077	−0.188

## Data Availability

The original contributions presented in this study are included in the article. Further inquiries can be directed to the corresponding author.

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
