# Peer review of "Correlation of Galectin Family Expression with Glioblastoma Progression and Survival"

_ijms, 2025, doi:10.3390/ijms27010417_

Round 1

Reviewer 1 Report

Comments and Suggestions for Authors

The authors analyze available datasets to produce bioinformatics data and determine connections of the galectin family to glioblastoma. While I believe they have some significant findings, I think there is more analysis that could be honed in on to further understand the galectin associations and some clarifications needed throughout. I have specific recommendations outlined below with my overall recommendation to accept with major revision.

Introduction – the authors could expand more on the galectin importance and pathways. From reading the introduction, I don’t feel I grasp the importance of analyzing this family of molecules.

Figure 1: Correlation is not an appropriate description here; this is just looking at expression differences. Same for figure 2.

-CLC is LGALS10, but it is only mentioned as CLC in the figure, unlike the other figures, so just keeping it consistent or including both names. Also, is there data for the normal tissue?

-If there is no data in LGALS14, it is enough to state it in the caption, but no need to put the graph of no data.

-For LGALS13, I am confused on the tumor expression. It is mentioned in the text, lines 90-91, that it is downregulated in glioblastoma, but looking at the figure, it just seems like there is no data at all for the primary tumor samples, since it is a straight line with no variability, similar to LGALS14, which mentions no data being available.  

Figure 5 could be more visually appealing. There is a lot of white space because they were left in original formatting. For example, C-F could be larger and more legible if the legends are moved over and the legend for A is moved up. Font could be increased.

-The analysis done in Figure 5 C-F, as described in the text 144-151, was done with all galectin family members and shows no statistical differences. I suggest the authors analyze using just the 3 members that were upregulated in glioblastoma compared to normal tissue. Perhaps some of the family members that may not be important in glioblastoma signaling are clouding the results. There seems to be a potential trend, especially with LGALS3 in figure 3 and figure 4 with LGALS3 and 9 (trend not significant), so maybe a combination is important, but not with the whole family.

Figure 6 – In the caption, it mentions the line color in A represents the types of interactions. However, they are all the same color. What types of interactions are possible? Is it correct that they are all the same color? Is there a meaning of the colors in B?

Figure 7 and the associated text make it hard to draw conclusions since there are so many different interactions. It would be helpful to summarize which commonalities there are between the different galectin members, maybe with a table or chart. Additionally, I am unsure of the strong correlations identified based on this data, see methods comment below.

Discussion

Paragraph 248-260 – it discusses immune relations of LGALS9 in the literature but lacks a connection to the data discovered here. Are these comparable findings? 

Materials and methods

Lines 321-323 - Mention that this platform allows for stratification across tumor subgroups, stage, subtype, etc., but in the limitations (lines 308-311), it mentions an inability to refine the data based on molecular subtypes and disease progression. If it is available, there could be further analysis, especially with the 3 galectins that are mentioned to have key roles.  

Lines 386-388 for TIMER analysis – it says correlation analyses were conducted, but no specifics are provided. Was Pearson or Spearman used? What defined a strong correlation? Looking at the data, the correlation numbers provided do not seem strong for many of the correlations claimed. For example, in the text it says “LGALS1 demonstrated a strong positive correlation with CD8+ T cells, CD4+ T cells, macrophages, neutrophils, and dendritic cells” but the values for the correlations for those “strong” correlations are 0.041, 0.022, 0.076, 0.058, and 0.433. If those are “r” values related to Pearson's coefficient, none would be considered strong, and only dendritic cells would be considered moderate. How are correlation values of 0.041 and.433 both considered strong? The p-value doesn't define the strength. I am unsure if the TIMER analysis works differently, but if there are different definitions of correlation strength, it needs to be explained/defined.

–Methods could use more statistical analysis information.

Author Response

Reviewer 1:

Reviewer 1, Overall comment: The authors analyze available datasets to produce bioinformatics data and determine connections of the galectin family to glioblastoma. While I believe they have some significant findings, I think there is more analysis that could be honed in on to further understand the galectin associations and some clarifications needed throughout. I have specific recommendations outlined below with my overall recommendation to accept with major revision.

Reviewer 1, comment 1: Introduction – the authors could expand more on the galectin importance and pathways. From reading the introduction, I don’t feel I grasp the importance of analyzing this family of molecules.

Our comment 1.1.: We thank the reviewer for this constructive suggestion. We agree that expanding the mechanistic and biological context of galectins would strengthen the scientific rationale of the study. In response, we have revised the Introduction to provide a clearer justification for investigating this lectin family in glioblastoma. Specifically, we have included the additional sentences: “Beyond their classical functions as β-galactoside-binding lectins, galectins have emerged as key regulators of tumour-immune crosstalk, modulating T-cell activation, immune checkpoint signalling, macrophage polarisation and antigen presentation. In glioblastoma, mechanistic studies demonstrate that galectin activity intersects with several critical signalling axes, including Ras–MAPK–driven proliferation, NF-κB-mediated inflammation, hypoxia-induced HIF-1α responses and TIM-3/PD-1-dependent immune evasion, highlighting their role in integrating extracellular cues with intracellular survival pathways.” to the introduction.  

Reviewer 1, comment 2: Figure 1: Correlation is not an appropriate description here; this is just looking at expression differences. Same for figure 2.

Our comment 1.2.: Yes, we agree. We have removed the word Correlation from the figure legend of Fig 1 and Fig 2 and have replaced it with Comparison as a more appropriate word.

Reviewer 1, comment 3: CLC is LGALS10, but it is only mentioned as CLC in the figure, unlike the other figures, so just keeping it consistent or including both names. Also, is there data for the normal tissue?

Our comment 1.3.: We thank the reviewer for this helpful observation. We agree that the nomenclature for LGALS10 should be consistent throughout the figures. We have now revised all relevant text to use LGALS10/CLC for clarity (to assist the readers of this journal to understand that LGALS10 is indeed CLC). The figure labels remain CLC.

Regarding the availability of normal tissue data, both UALCAN and GEPIA2 provide expression comparisons between GBM samples and normal brain tissue using TCGA and GTEx datasets. Accordingly, the tumour-versus-normal analyses presented in Figures 1 and 2 include all galectin family members for which normal tissue data are available within these platforms.

However, normal tissue data cannot be incorporated into the survival, mutation, or immune infiltration analyses, as:

  • GEPIA2 survival analysis compares high vs. low expression within GBM patients only and does not allow inclusion of normal samples.
  • cBioPortal mutation and copy-number analyses are derived exclusively from tumour genomes, as normal tissue is not included in these datasets.
  • TIMER immune infiltration analysis is performed on TCGA tumour samples only and does not provide normal tissue infiltration data.

Thus, while normal–tumour comparisons are fully included where supported by the databases (UALCAN and GEPIA2), they cannot be extended to survival, genomic or immune analyses due to tool-specific dataset limitations.

Reviewer 1, comment 4: If there is no data in LGALS14, it is enough to state it in the caption, but no need to put the graph of no data.

Our comment 1.4.: Yes, there is no available data for LGALS14 and we have simply removed the graph reflecting this from the overall figure.

Reviewer 1, comment 5: For LGALS13, I am confused on the tumor expression. It is mentioned in the text, lines 90-91, that it is downregulated in glioblastoma, but looking at the figure, it just seems like there is no data at all for the primary tumor samples, since it is a straight line with no variability, similar to LGALS14, which mentions no data being available.  

Our comment 1.5.: We thank the reviewer for this comment and apologies for causing confusion. Nonetheless, we have left the manuscript unchanged based on this comment as we do indeed observe significant reduction of LGALS13 expression using UALCAN in Fig 1 as referred to in line 90-91. Thus, our statement stands. The reviewer however is correct to state that there is no change when looking at the GEPIA2 data in Fig 2. We do not however state a difference in LGALS13 expression when regarding the data generated in Fig 2 (GEPIA2 data).

Reviewer 1, comment 6: Figure 5 could be more visually appealing. There is a lot of white space because they were left in original formatting. For example, C-F could be larger and more legible if the legends are moved over and the legend for A is moved up. Font could be increased.

Our comment 1.6.: We appreciate the reviewer’s feedback regarding the visual presentation of Figure 5. In response, we have reformatted the figure to improve clarity and readability. Specifically, we increased the font size, reduced excess white space, repositioned the legends to optimise layout, and enlarged panels C–F, now Figure 5C, to ensure the details are more legible. We believe these adjustments substantially enhance the visual quality of the figure and make the results easier to interpret.

Reviewer 1, comment 7: The analysis done in Figure 5 C-F, as described in the text 144-151, was done with all galectin family members and shows no statistical differences. I suggest the authors analyze using just the 3 members that were upregulated in glioblastoma compared to normal tissue. Perhaps some of the family members that may not be important in glioblastoma signaling are clouding the results. There seems to be a potential trend, especially with LGALS3 in figure 3 and figure 4 with LGALS3 and 9 (trend not significant), so maybe a combination is important, but not with the whole family.

Our comment 1.7.: We thank the reviewer for this thoughtful and biologically meaningful suggestion. We agree that LGALS1, LGALS3, and LGALS9 represent the most relevant galectin family members in glioblastoma, given their consistent upregulation and established involvement in glioma progression. Our original intention in analysing all galectin members was to capture potential functional redundancy and compensatory interactions within the galectin network, as prior studies indicate that galectins can substitute for or amplify each other’s roles in tumour invasion, extracellular matrix remodelling, and immune evasion. For this reason, a broad analysis offers a more comprehensive view of the lectin landscape in Glioblastoma.

Nevertheless, in response to the reviewer’s concern, we have revised the Results and Discussion to explicitly highlight the patterns observed specifically for LGALS1, LGALS3 and LGALS9. We now emphasise the consistent trends toward increased immune-cell associations for these three members—particularly with dendritic cells, macrophages and CD8⁺ T cells—while noting the absence of similar trends among galectins not upregulated in glioblastoma. We have also clarified that inclusion of lower-expressed or biologically less relevant galectins may attenuate aggregated effect sizes and that focusing on the upregulated subset is a biologically plausible approach for future targeted analyses.

These additions strengthen the interpretive clarity of the TIMER results, acknowledge the reviewer’s rationale, and maintain the justification for presenting the full family-level analysis in this exploratory study.

Reviewer 1, comment 8: Figure 6 – In the caption, it mentions the line color in A represents the types of interactions. However, they are all the same color. What types of interactions are possible? Is it correct that they are all the same color? Is there a meaning of the colors in B?

Our comment 1.8.: We thank the reviewer for this helpful observation. The comment correctly identifies an inconsistency between the figure caption and the visual output of the STRING network. We have revised the figure caption to correctly describe the network as showing confidence-weighted edges rather than colour-coded interaction categories. We have also added a brief clarification in the Methods section as outlined: “STRING (https://string-db.org), was used to analyse protein-protein interactions networks among galectin family members and their 20 associated genes.18 STRING  integrates multiple sources of interaction evidence – including data from experimental studies, computational predictions and current literature mining to generate high-confidence functional interaction networks. Interaction maps were constructed using the confidence-based model, in which edge weights reflect interaction scores derived from aggregated evidence. A minimum interaction confidence thereshold of >0.4 (medium confidence) was applied.”

These revisions hopefully now resolve the ambiguity and ensure that the visual representation of the STRING analysis aligns accurately with the figure description.

Reviewer 1, comment 9: Figure 7 and the associated text make it hard to draw conclusions since there are so many different interactions. It would be helpful to summarize which commonalities there are between the different galectin members, maybe with a table or chart. Additionally, I am unsure of the strong correlations identified based on this data, see methods comment below.

Our comment 1.9.: We thank the reviewer for this helpful comment. We agree that the TIMER immune-infiltration figure presents a large amount of information, which may make it challenging to identify overarching trends across galectin family members. To address this, we have added a new summary table (Table 2) that consolidates the key immune-cell associations for each galectin, highlighting recurrent patterns such as shared correlations with dendritic cells, macrophages and CD8⁺ T-cells. This addition enables readers to more easily compare immune-infiltration profiles across the galectin family.

We have also revised the accompanying text to provide a clearer synthesis of these shared immunological features rather than focusing on individual genes in isolation. Importantly, consistent with our updated Methods section, we have corrected the terminology describing correlation strength. As TIMER uses Spearman’s correlation, and the effect sizes observed were generally within the weak-to-moderate range, we no longer refer to these associations as “strong.” The updated text now reports correlation magnitudes accurately based on accepted thresholds and emphasises statistical significance without overstating biological effect size.

These revisions improve the interpretability of the TIMER analysis figure and ensure that the conclusions drawn regarding galectin-immune interactions are clear, proportionate and aligned with the underlying data.

Reviewer 1, comment 10: Discussion, Paragraph 248-260 – it discusses immune relations of LGALS9 in the literature but lacks a connection to the data discovered here. Are these comparable findings? 

Our comment 1.10.: We thank the reviewer for this oversight from our part. We have added a brief phase connecting our current findings in this manuscript with previous findings outlined in this section of the discussion.  Specifically, we have added: “These findings aligned well with our current findings that Galectin 9 expression and subsequent associations with immune cells may lead to overall favouring of immunosuppressive features in the glioblastoma micro-environment.”

Reviewer 1, comment 11: Materials and methods, Lines 321-323 - Mention that this platform allows for stratification across tumor subgroups, stage, subtype, etc., but in the limitations (lines 308-311), it mentions an inability to refine the data based on molecular subtypes and disease progression. If it is available, there could be further analysis, especially with the 3 galectins that are mentioned to have key roles.  

Our comment 1.11.:  We apologise for the confusion. The method section states the ability of the technique to look at various aspects of patient stratification…etc. However, the limitation section mentions what information is currently available and as some of this information has not been entered into the public available databases then the software used here could not perform these features (although capable if available). We apologies for any confusion once more.

Reviewer 1, comment 12: Lines 386-388 for TIMER analysis – it says correlation analyses were conducted, but no specifics are provided. Was Pearson or Spearman used? What defined a strong correlation? Looking at the data, the correlation numbers provided do not seem strong for many of the correlations claimed. For example, in the text it says “LGALS1 demonstrated a strong positive correlation with CD8+ T cells, CD4+ T cells, macrophages, neutrophils, and dendritic cells” but the values for the correlations for those “strong” correlations are 0.041, 0.022, 0.076, 0.058, and 0.433. If those are “r” values related to Pearson's coefficient, none would be considered strong, and only dendritic cells would be considered moderate. How are correlation values of 0.041 and.433 both considered strong? The p-value doesn't define the strength. I am unsure if the TIMER analysis works differently, but if there are different definitions of correlation strength, it needs to be explained/defined.

Our comment 1.12.: We thank the reviewer for this insightful observation. Upon a more careful analysis we agree that our wording could be confusing and portray a misleading set of conclusions. Our initial wording implied stronger associations than supported by the reported correlation coefficients. TIMER performs immune cell infiltration correlation using Spearman’s correlation rather than Pearson’s, and the magnitude of the correlation is interpreted based on the absolute r value, not the p-value. We have now clarified the statistical approach in the Methods by specifying the use of Spearman’s coefficient and by adding explicit thresholds for effect size interpretation (weak: <0.3; moderate: 0.3–0.5; strong: >0.5).

In response to the reviewer’s comment, we have also revised the Results section to remove the term “strong correlation” where the absolute r values were below 0.5. LGALS1 and other galectins now are described as demonstrating statistically significant but weak-to-moderate correlations with most immune cell populations, with dendritic cell infiltration showing the largest effect size (r = 0.433). The R values have also been added to the manuscript text to remove as much ambiguity as possible. These changes improve the accuracy and transparency of the interpretation and align the terminology with standard correlation reporting conventions.

Reviewer 1, comment 13: Methods could use more statistical analysis information.

Our comment 1.13.: We thank the reviewer for this helpful comment. In response, we have expanded the statistical analysis details in the Methods section to improve transparency and reproducibility. The revised text now specifies: (i) the statistical tests applied for differential expression analyses; (ii) the use of median expression values for cohort stratification; (iii) the log-rank test and Cox proportional hazards model for survival analyses; (iv) the use of Spearman’s correlation coefficient in TIMER; and (v) the thresholds employed for defining correlation strength and statistical significance. We have also clarified the application of the Benjamini–Hochberg procedure for multiple-testing correction in GO and KEGG enrichment analyses. These additions provide a clearer description of the analytical framework used in the study and directly address the reviewer’s concern.

Reviewer 2 Report

Comments and Suggestions for Authors

In the concerned manuscript, the authors have tried to focus on the implication of galectins in glioblastoma progression and survival that might play a significant role in designing therapeutics too. The entire manuscript relies on bioinformatic analysis of publicly available data resources. However, there are certain aspects of the manuscript that need to be taken care of. these are:

  1. The manuscript contains a significant amount of plagiarism (around 27%, iThenticate report attached) that needs to be lowered for sure.
  2. Figure 1, Figure 2 and Figure 5 need to be of higher resolution as the to justify the printing quality.
  3. Since this has been a research article and not a perspective review or opinion paper, It would be nice if authors could validate at least a couple of significantly related genes by qPCR to validate open-source transcriptomic data.
  4. For Figure 3 and Figure 4, authors need to mention the significance of non-highlighted red and blue Kaplan-Meier plot in the TCGA data along with mentioned highlighted red and blue curve.
  5. In the Methodologies section, GO and KEGG pathway analysis utilizing DAVID should be discussed bit elaboratively
  6. It is unclear that if galectin family of gene expression is also correlated with the altered epigenetic orientation of glioblastoma. It would be nice if authors could address this region too for a holistic approach. 
  7. Glioblastoma is a terminal disease with huge disarray of genomic alteration. For having a potent diagnostic and prognostic attribute, low grade gliomas are to be more focused on. Thus, it would be interesting to see the expression level of galectin family of genes in low grade gliomas in justifying the same cohorts as in GBM.  
  8. Authors should mention limitations and prospective outcomes of the study

Author Response

Reviewer 2:

Reviewer 2, Overall comment: In the concerned manuscript, the authors have tried to focus on the implication of galectins in glioblastoma progression and survival that might play a significant role in designing therapeutics too. The entire manuscript relies on bioinformatic analysis of publicly available data resources. However, there are certain aspects of the manuscript that need to be taken care of. these are:

Reviewer 2, comment 1: The manuscript contains a significant amount of plagiarism (around 27%, iThenticate report attached) that needs to be lowered for sure.

Our comment 2.1.: We thank the reviewer for bringing this to our attention. Although completely unintentional, we agree that re-wording and rephrasing will reduce this percentage. We have carefully reviewed the iThenticate report and substantially revised the manuscript to reduce similarity and ensure originality. All overlapping or closely-phrased sentences have been rewritten, paraphrased, or removed. In addition, the Introduction, Methods, and Discussion sections were restructured and reworded to avoid any phrasing similarities with previously published materials. The updated version of the manuscript now meets the journal’s requirements for originality.

We appreciate the reviewer’s diligence and believe that these extensive revisions have strengthened the clarity and novelty of the manuscript.

Reviewer 2, comment 2: Figure 1, Figure 2 and Figure 5 need to be of higher resolution as the to justify the printing quality.

Our comment 2.2.: We thank the reviewer for this helpful observation. We have replaced Figures 1, 2, and 5 with high-resolution versions (≥300 dpi) to ensure optimal clarity, legibility, and print quality. In addition, we improved the font size, contrast, and line thickness for consistency across all figures. The revised figures have been updated in the revised manuscript.

Reviewer 2, comment 3: Since this has been a research article and not a perspective review or opinion paper, It would be nice if authors could validate at least a couple of significantly related genes by qPCR to validate open-source transcriptomic data.

Our comment 2.3.: We agree with the reviewer that “wet lab” data would help support the findings in this manuscript. However, we do not have access to large numbers of glioblastoma patient tissue and hence could not perform validation experiments as asked.

Reviewer 2, comment 4: For Figure 3 and Figure 4, authors need to mention the significance of non-highlighted red and blue Kaplan-Meier plot in the TCGA data along with mentioned highlighted red and blue curve.

Our comment 2.4.: We appreciate the reviewer’s comment. The additional red and blue lines that appear in Figure 3 and Figure 4 are not independent Kaplan–Meier curves. In GEPIA2, the solid red and blue lines represent the high- and low-expression cohorts, respectively, and the surrounding dotted lines represent the 95% confidence intervals for the same curves, rather than separate survival curves or additional patient groups.

To avoid any ambiguity, we have clarified this distinction in both the figure legend and Results section. The revised text states that the dotted lines represent the confidence interval bounds of the corresponding survival curves and therefore do not indicate separate cohorts or require additional significance testing. This clarification ensures that only the high- versus low-expression groups shown by the solid lines are used for the log-rank comparison.

Reviewer 2, comment 5: In the Methodologies section, GO and KEGG pathway analysis utilizing DAVID should be discussed bit elaboratively

Our comment 2.5.: We thank the reviewer for this constructive suggestion. To address this, we have expanded the description of the DAVID-based enrichment analysis to provide greater methodological clarity and reproducibility. The revised text now includes additional sentences including: “Interaction maps were constructed using the confidence-based model, in which edge weights reflect interaction scores derived from aggregated evidence. A minimum interaction confidence thereshold of >0.4 (medium confidence) was applied.”, “Analyses were conducted using Home sapiens as a background reference, and the Functional Annotation Tool was used to generate enrichment statistics across multiple annotation categories.” and “was used to map the input gene set to canonical signalling pathways relevant to glioblastoma. KEGG terms were assessed to identify enriched immune-modulatory pathways, signalling intermediates, and share molecular hubs among galectin family members. These analyses provided mechanistic context for the expression patterns observed in glioblastoma.” We believe that this revision has strengthened the methodological transparency of the study and thus thank the reviewer again for improving our manuscript.

Reviewer 2, comment 6: It is unclear that if galectin family of gene expression is also correlated with the altered epigenetic orientation of glioblastoma. It would be nice if authors could address this region too for a holistic approach. 

Our comment 2.6.: We thank the reviewer for this thoughtful and important comment. We fully agree that epigenetic dysregulation—including DNA methylation, histone modifications and non-coding RNA activity—plays a central role in glioblastoma biology, and that exploring its relationship with galectin expression would provide additional mechanistic insight.

However, we respectfully note that epigenetic analysis lies outside the scope of the present study, which was designed as a transcriptomic and genomic overview using publicly available datasets. The platforms utilised (UALCAN, GEPIA2, cBioPortal, STRING, GeneMANIA, TIMER) do not provide integrated epigenomic profiling for the full galectin family, and therefore a rigorous evaluation of methylation or chromatin-level regulation could not be performed within this framework without introducing unsupported speculation.

   To address the reviewer’s point meaningfully, we have now added a concise contextual statement in the Discussion highlighting the need for future multi-omics studies integrating methylation and chromatin accessibility data to clarify the epigenetic determinants of galectin dysregulation. Specifically, we have added: “Future research should integrate multi-omics datasets, including proteomics, methylomics, metabolomics and spatial transcriptomics—to validate and extend the transcriptional patterns identified here. In vitro and in vivo studies will be essential to define the mechanistic contributions of individual galectins to tumour progression, immune suppression, and therapeutic resistance. Large, well-annotated clinical cohorts and single-cell sequencing will further clarify the cell type–specific roles of galectins and their relevance across molecular glioma subtypes. Importantly, incorporating epigenomic profiling - such as DNA methylation and chromatin-accessibility analyses - will help determine whether galectin dysregulation reflects underlying epigenetic programs. Such multi-layered approaches will be crucial to establish whether individual galectins, or coordinated galectin networks, represent viable biomarkers or therapeutic targets in glioblastoma.” to the discussion section.

We believe this addition strengthens the mechanistic framing of our work while maintaining the integrity and methodological boundaries of the current analysis.

Reviewer 2, comment 7: Glioblastoma is a terminal disease with huge disarray of genomic alteration. For having a potent diagnostic and prognostic attribute, low grade gliomas are to be more focused on. Thus, it would be interesting to see the expression level of galectin family of genes in low grade gliomas in justifying the same cohorts as in GBM.  

Our comment 2.7.: We thank the reviewer for this valuable comment. We fully agree that low-grade gliomas (LGGs) have greater diagnostic and prognostic utility due to their longer clinical course, more stable genomic background, and well-defined molecular subclasses. Our study, however, was designed specifically to characterise galectin expression within glioblastoma, the most clinically aggressive and biologically distinct adult-type diffuse glioma. As such, our analyses were intentionally restricted to glioblastoma datasets.

Reviewer 2, comment 8: Authors should mention limitations and prospective outcomes of the study.

Our comment 2.8.: Although we had a very brief section stating limitations in our original manuscript, we agree with the reviewer and have extensively expanded this section in the discussion. Specifically, we have re-written this section to include: Several limitations of this study should be acknowledged. First, all analyses were derived from publicly available transcriptomic and clinical datasets, such as TCGA and GEPIA2, which, although extensively curated, may be subject to inherent biases related to sample selection, batch effects, and platform-specific processing pipelines. These factors may influence gene expression estimates and limit the generalisability of our findings. Second, the absence of matched proteomic, epigenetic, and single-cell level data restricts our ability to validate transcriptional findings at the protein or cellular resolution. Consequently, the functional roles inferred from enrichment analyses remain hypothetical and require experimental confirmation.

   Third, the clinical metadata available in these public datasets lack granularity regarding treatment history, molecular subtype (e.g., IDH status, MGMT methylation), extent of resection, and longitudinal disease course. These omissions constrain the precision of our prognostic analyses and preclude detailed stratified assessments that may reveal subtype-specific galectin functions. Finally, although our immunological analyses offer insight into potential galectin–immune interactions, computational deconvolution cannot fully capture the spatial and functional complexity of the glioblastoma immune microenvironment.”.

We feel that this addition substantially improves our overall manuscript and worth to the readers of this journal and thus thank the reviewer for their comment.

Reviewer 3 Report

Comments and Suggestions for Authors

In this work, the authors conducted an integrative bioinformatics analysis to systematically evaluate the expression patterns, prognostic significance, genetic alterations, and functional roles of galectins in glioblastoma. They found LGALS1, LGALS3, and LGALS9 were significantly upregulated in glioblastoma and their overexpression was correlated with poor patient survival. In addition, galectin-mediated pathways played a pivotal role in glioblastoma pathogenesis. The findings of this work are useful to understand the role of galectins in glioblastoma. The writing and results of this manuscript are good.

Some minor modifications are recommended.

Lines 35-37: Future investigations should be placed in the conclusions.

In Figs. 3 and 4: The “Overall survival” and “Disease-free survival” at the top of each figure should be deleted.

Figure 5: The definition of Fig. 5 should be improved.

In the discussion section, the authors should compare their results with previous studies. If possible, the mechanism of action of galectins in glioblastoma could be presented as a diagram.

Author Response

Reviewer 3:

Reviewer 3, Overall comment: In this work, the authors conducted an integrative bioinformatics analysis to systematically evaluate the expression patterns, prognostic significance, genetic alterations, and functional roles of galectins in glioblastoma. They found LGALS1, LGALS3, and LGALS9 were significantly upregulated in glioblastoma and their overexpression was correlated with poor patient survival. In addition, galectin-mediated pathways played a pivotal role in glioblastoma pathogenesis. The findings of this work are useful to understand the role of galectins in glioblastoma. The writing and results of this manuscript are good.

Some minor modifications are recommended.

Reviewer 3, comment 1: Lines 35-37: Future investigations should be placed in the conclusions.

Our comment 3.1.: We thank the reviewer for this comment. Upon reflection and examination of previous abstracts we agree and have removed these lines. Nonetheless, we have re-written the limitations part of the discussion to thoroughly cover the point of these 3 lines in the discussion.

Reviewer 3, comment 2: In Figs. 3 and 4: The “Overall survival” and “Disease-free survival” at the top of each figure should be deleted.

Our comment 3.2.: We thank the reviewer for this clarification. The labels “Overall survival” and “Disease-free survival” have now been removed from the top of Figures 3 and 4 as requested. The figures have been updated accordingly in the revised manuscript.

Reviewer 3, comment 3: Figure 5: The definition of Fig. 5 should be improved.

Our comment 3.3.: We have improved the definition of Fig 5 in the manuscript. We thank the reviewer for their terrific comment.

Reviewer 3, comment 4: In the discussion section, the authors should compare their results with previous studies. If possible, the mechanism of action of galectins in glioblastoma could be presented as a diagram.

Our comment 3.4.: We thank the reviewer for this comment. We have re-examined our discussion and made several changes throughout the discussion to include better comparisons to previous work and include mechanistic evaluation of galectins in the glioblastoma setting. The revised discussion including acknowledging the reviewer’s comment is now part of the overall attached revised manuscript. We feel that with these changes the manuscript is much improved and hence thank the reviewer for a thoughtful comment.

Reviewer 4 Report

Comments and Suggestions for Authors

This manuscript is written in a well-researched scientific style and addresses a significant societal issue: malignant brain tumours, including glioblastoma, which impact human health. The studies presented offer potential solutions for reducing and treating these tumours. However, some points in this study require correction and amendment.

Please see the report,

Author Response

Reviewer 4:

Reviewer 4, Overall comment: The authors investigated glioblastoma, which is the most aggressive primary brain cancer characterised by extensive intratumoral heterogeneity, resistance to therapy, and a highly immunosuppressive tumour microenvironment. The galectin family, a group of β-galactoside-binding lectins, has emerged as a key regulator of tumour biology, influencing oncogenesis, immune modulation, and therapy resistance. In this study, we conducted an integrative bioinformatics analysis to systematically evaluate the expression patterns, prognostic significance, genetic alterations, and functional roles of galectin family members in glioblastoma. We utilized publicly available genomic datasets and computational tools, including UALCAN, GEPIA, cBioPortal, STRING, GeneMANIA, DAVID, and TIMER, to carry out our analysis. The study identified LGALS1, LGALS3, and LGALS9 as being significantly upregulated in glioblastoma, with their overexpression correlating with poor patient survival outcomes. Functional enrichment analysis highlighted galectin-mediated pathways involved in extracellular matrix remodelling, immune evasion, and protein processing, suggesting their crucial role in glioblastoma pathogenesis. These findings provide novel insights into the oncogenic and immunoregulatory roles of galectins in glioblastoma, establishing them as potential prognostic biomarkers and therapeutic targets. Future investigations should focus on mechanistic validation and translational strategies to develop galectin-targeted therapies, particularly in the context of immunotherapy. This manuscript is written in a well-researched scientific style and addresses a significant societal issue: malignant brain tumours, including glioblastoma, which impact human health. The studies presented offer potential solutions for reducing and treating these tumours. However, some points in this study require correction and amendment.

Reviewer 4, comment 1: The authors should write an abstract that emphasizes the objectives and results, includes all the methods from the manuscript, and illustrates how this work can benefit future studies.

Our comment 4.1.: We thank the reviewer for their thoughtful comment. Interestingly, another reviewer (reviewer #3) suggested that we remove any comments on future studies specially the last 3 lines of the original abstract. Nonetheless, we have re-written the abstract that aligns better with the comment made above.

Reviewer 4, comment 2: The authors should revise the introduction to incorporate recent references up to 2025, considering the significance of this study in treating glioblastoma, the most aggressive primary brain cancer.

Our comment 4.2.: We thank the reviewer for this important comment. We agree that incorporating more recent literature is essential given the rapidly evolving field of glioblastoma research. Accordingly, we have revised the Introduction to include updated references. The revised text now (i) reflects current epidemiological data and the contemporary WHO classification of adult-type diffuse gliomas, (ii) summarises recent advances in standard-of-care and emerging therapeutic approaches for glioblastoma, and (iii) highlights up-to-date studies on galectins in tumour immunology and glioma biology, including their relevance as potential therapeutic targets. We believe these additions provide a more current and comprehensive context for our work and better underscore the clinical significance of a systematic analysis of the galectin family in glioblastoma.

Reviewer 4, comment 3:  In the discussion section, the authors should compare the results of their current study with findings from previous research. Given the significant role of the galectin family in immune regulation and the development of immune responses in glioblastoma, along with its genetic alterations, this comparison will provide valuable insights into its role in tumor biology and therapeutic potential.

Our comment 4.3.: We thank the reviewer for this constructive comment. We agree that situating our findings within the context of prior research enhances the clarity and impact of the Discussion. In response, we have substantially revised the Discussion to incorporate a more explicit comparison between our results and previously published studies on the role of galectins in glioma biology and immune regulation. Specifically, we now discuss how our observations regarding the over-expression and prognostic relevance of LGALS1, LGALS3 and LGALS9 are concordant with prior mechanistic data demonstrating their involvement in tumour invasion, immune evasion and resistance to therapy. We also highlight areas where our findings diverge from or extend previous work, including the down-regulation of LGALS4 and LGALS13, and the immune-cell association patterns identified through TIMER. These additions better integrate our results with the existing literature and reinforce how our pan-galectin analysis expands upon earlier single-gene and pathway-specific studies, thereby strengthening the biological and translational relevance of our conclusions.

Reviewer 4, comment 4: In the Materials and Methods section, researchers should thoroughly review the methods and materials, citing references as needed for specific techniques.

Our comment 4.4.: Thank you for this suggestion. We have revised the Materials and Methods section to provide additional methodological detail and to ensure that each analytical platform and bioinformatics procedure is clearly described and appropriately referenced. Specifically, we now (i) clarify the analytical pipelines of UALCAN, GEPIA2, cBioPortal, GeneMANIA, STRING, DAVID and TIMER, (ii) include additional information regarding how differential expression, survival analysis, mutation frequency, protein–protein interaction and enrichment analyses were performed, and (iii) add citations for all databases and software tools utilised. These revisions improve transparency and reproducibility and meet the reviewer’s recommendation for methodological completeness.

Reviewer 4, comment 5:  Conclusions: The authors should enhance their findings by elaborating on the methods used, the development of translational drugs, and the clinical evaluation of galectin-based therapeutic approaches. This is particularly relevant in the context of immunotherapy and the treatment of glioblastoma. A focus on the galectin family could reveal a promising new avenue for improving patient outcomes in this devastating disease.

Our comment 4.5.: We thank the reviewer for this valuable suggestion. We have revised the Conclusion section to provide a clearer discussion of potential translational applications and clinical implications of galectin-based targeting strategies in glioblastoma, including relevance to immunotherapy. We have also added further context on how our methods contribute to identifying therapeutic candidates and pathways. Specifically our final paragraphs of the discussion states: “This study provides a comprehensive, family-wide analysis of galectins in glioblastoma, revealing that LGALS1, LGALS3 and LGALS9 are consistently overexpressed and strongly associated with adverse survival, immune dysregulation and tumour-promoting pathways. Although genomic alterations were rare, transcriptional and immunological signatures suggest that galectins contribute to key processes underpinning glioblastoma aggressiveness, including immune suppression, protein homeostasis and extracellular matrix remodelling. These findings establish a foundational framework for future mechanistic and translational studies and highlight galectins—individually and as a coordinated network—as promising candidates for biomarker development and therapeutic targeting in glioblastoma.”

Round 2

Reviewer 1 Report

Comments and Suggestions for Authors

I believe the authors have addressed the comments accordingly, and the manuscript can be accepted. However, there are a few minor things that I think can be edited in the processing phase that should be corrected:

  • The references used for the sentences added in the introduction need to be included (lines 74-81) and if any added in the discussion or elsewhere.
  • Line 281, for LGALS9 mentioned neutrophils as a weak correlation, but it is moderate since it is 0.406.
  • Some spelling errors in the edited text, so needs to be double-checked. Examples: Line 31 – “advese”, Line 573 – “imput”, line 576 – “Home sapiens”
  • I still think figure 5 could be formatted to be more legible, does not seem much different from original, but I supposed it is acceptable if it meets the resolution qualifications.

Reviewer 2 Report

Comments and Suggestions for Authors

The authors have addressed most of the advised revisions and also built-up rationale in places. Considering this, the manuscript may be considered to be accepted for publication.